# The experiences of high-risk young adults with type 1 diabetes transitioning to real-time continuous glucose monitoring – A thematic analysis

Vicky McKechnie[1], Parizad Avari[1], Pei Chia Eng[1,2], Rebecca Unsworth[1], Monika Reddy[1], Stephanie A. Amiel[3], Victoria Salem[4], Shivani Misra[1*]

1 Department of Metabolism, Digestion and Reproduction, Imperial College London, London, United Kingdom, 2 Department of Endocrinology, National University of Singapore, Singapore, Singapore, 3 Department of Diabetes, Faculty of Life Sciences, King's College London, London, United Kingdom, 4 Department of Bioengineering, Imperial College London, London, United Kingdom

* s.misra@imperial.ac.uk

## Abstract

### Background

Real-time continuous glucose monitoring (rtCGM) is now the standard care for people with type 1 diabetes. However, whilst its impact on glycaemic outcomes is well-documented, its psychosocial effects, particularly in young adults experiencing extreme hyperglycaemia, remain poorly understood.

### Objectives

We aimed to explore the psychosocial impact of rtCGM on young adults with extreme hyperglycaemia who thus far have not been studied extensively.

### Research design and methods

A qualitative study employing semi-structured interviews was undertaken. Young adults 18–25 years (HbA1c >75mmol/mol (9.0%)), naïve to rtCGM, were provided with rtCGM for 6-months. Interviews (centred on barriers to self-management and experience of rtCGM use) were conducted within 2-weeks of recruitment and at the end. An inductive, thematic analysis of interviews was undertaken.

### Results

Eight participants (median age (IQR) 23.0 (22.0–24.5) years, 100% non-white ethnicity) were recruited with median HbA1c 94 (88–107) mmol/mol [DCCT 10.8 (10.2–12.1)%.]. All participants used multiple daily insulin injections. Despite low rtCGM wear-time (32.2 (23.1–59.4)%), significant improvements were observed in time in range, but no change in HbA1c. Thematic analysis indicated that high levels of

**Data availability statement:** All relevant data are within the manuscript and its Supporting Information files. Further data cannot be shared publicly because of risk participant identification. However, reasonable requests for further data can be sent to Dr Lalantha Leelarathna, Clinical Senior Lecturer in Diabetes (e.leelarathna@imperial.ac.uk) for consideration.

**Funding:** Dexcom provided funding for sensors and research staff time. The funder of the study had no role in study design, data collection, data analysis, data interpretation, or writing of the report. SM is funded by a Wellcome Trust Career Development Award (223024/Z/21/Z) and is supported by the NIHR Imperial Biomedical Research Centre. PA is funded by NIHR Grant No. 132960. VS is a Diabetes UK Grand Challenge Senior Fellow as is also supported by grants currently held from the MRC, BBSRC and NC3Rs. PCE is supported by the NUS Exxon Mobil Grant and the National Medical Research Council (NMRC) New Investigator Grant, Singapore.

**Competing interests:** I have read the journal's policy and the authors of this manuscript have the following competing interests: SM served as Trustee to the Diabetes Research & Wellness Foundation charity, UK until 2024 and has received speaker Honoraria from Lilly and Sanofi, UK. MR has received honoraria for advisory board participation from Dexcom and Roche Diabetes. SAA has served on Advisory Boards for Novo Nordisk and Medtronic and has spoken at educational meetings supported by NovoNordisk and Sanofi. PA has received equipment from Dexcom for investigator-initiated studies. VM, PCE, RU have no conflicts of interest to declare. I confirm that none of the authors' competing interests alter our adherence to PLOS ONE policies on sharing data and materials.

disease burden were reported, with rtCGM-related themes identified: 1) interaction with rtCGM data, 2) feelings of control and trust from using rtCGM, and 3) frustration of technology and alarms. Although participants reported that knowledge of glucose levels on their smartphone was convenient and led to 'greater control', this was countered by alarm-fatigue, technical difficulties and feeling overwhelmed. Three participants prematurely stopped using rtCGM.

## Conclusions

Young adults with high-risk hyperglycaemia have complex relationships with rtCGM. rtCGM may have benefits in this high-risk group, but are likely to require additional support and must be determined on a case-by-case basis as associated effort may contribute to feelings of distress and/or burnout. Implementing structured educational, psychosocial, and technical support, alongside alternative care models such as more frequent check-ins, should be considered in order to enhance self-management practices with rtCGM and address technology-related challenges.

## Introduction

Intensive self-management of type 1 diabetes improves glucose control and reduces the risk of microvascular complications and cardiovascular disease [1]. The young-adult population with type 1 diabetes are particularly susceptible to barriers impeding optimal self-management, which may be cognitive, emotional and/or practical in nature [2]. These include heightened concerns about peer relationships and social interactions, fatigue from chronic-disease management, a transition in responsibility of management from parents and an inclination towards risk-taking [2]. Young adults may face additional life challenges such as being in full-time education, starting employment and moving to independent living [3,4]. The multiple factors challenging self-management in younger people with type 1 diabetes is reflected in their observed high risk for acute hyperglycaemic crises such as diabetic ketoacidosis (DKA) and recurrent hospital admissions for hyperglycaemia [5] and in higher levels of glycated haemoglobin (HbA1c) [6,7].

Real-time continuous glucose monitoring (rtCGM) is associated with lower HbA1c levels [8–11], and benefits are linked to wear-time [12]; in the JDRF-CGM randomized controlled trial (RCT), [13], participants aged >25-years experienced a reduction in HbA1c associated with sensor wear-time, however no differences in HbA1c were observed in 14–25-year-olds with lowest sensor-usage. These results were confirmed in other studies [14].

A more contemporary RCT in adults aged 16–24-years (MILLENIALS) demonstrated percentage time in range was significantly higher in users of rtCGM versus finger-prick testing [15], and sensor wear-time was higher than observed in older RCTs, possibly reflecting improvements in technology performance and changing attitudes towards wearable technology.

Many rtCGM studies have excluded people with extremely high HbA1c due to safety concerns; the mean HbA1c (14–25-year-olds) in the JDRF-CGM study was 63.9mmol/mol (8%) [13] and 78.4mmol/mol (9.3%) in the MILLENNIALS study [15]. Individuals with much higher HbA1c are likely to be those at highest risk of hyperglycaemic emergencies and insulin omission and are also individuals who may struggle with self-management due to other factors such as mental well-being or burnout.

There is limited qualitative research looking into the views and experiences of rtCGM usage in high-risk young adults with type 1 diabetes and there is a research gap concerning the psychosocial effects of rtCGM for this high-risk group. It is therefore unknown how rtCGM may affect thoughts and behaviours that are barriers to achieving optimum or safe glycaemia in this group. There is a clinical need to identify strategies that can address barriers in this vulnerable group. In this study, we assessed the experiences and impact of rtCGM in very high-risk individuals with suboptimal HbA1c at baseline.

This study addresses two key research questions: (1) How do individuals with high-risk T1D experience and manage life with diabetes, including barriers to self-management? (2) How do participants perceive and use rtCGM, including challenges and its impact on diabetes management?

## Methods

Qualitative data were collected as part of the Real-time Continuous Glucose Monitoring in High-risk Young Adults with Insulin-treated Diabetes (YODA) study. YODA aimed to analyze the effects of 6 months of rtCGM in a randomised controlled crossover study against standard care. The study was terminated early due to new guidance from National Institute for Health and Care Excellence (NICE) published in 2022 [16] advocating rtCGM for all adults with type 1 diabetes as standard care. For the present analysis, we report the psychosocial outcomes from a thematic analysis of interviews pre- and post-rtCGM. For context, we have included CGM metrics and psychosocial questionnaire data from the intervention group.

The study was approved by the UK Health Research Authority London Bridge Research Ethics Committee (Reference number 19/LO/1341; ClinicalTrials.gov NCT04039763). All participants provided verbal and written informed consent.

### Participants

Young adults aged 18–25 years with insulin-treated diabetes >12 months (on multiple daily injections or insulin pump therapy) were included if they demonstrated one or more of: HbA1c > 75 mmol/mol (9%) or ≥ 1 DKA admissions or ≥ 1 hospital admission with extreme hyperglycaemia in the last 18 months. These criteria were based on people with recurrent DKA having higher HbA1c [17] and readmission rates [18]. Many early rtCGM studies have excluded people with extremely high HbA1c due risks of DKA.

Individuals were naïve to rtCGM and previous or current use of intermittently-scanned CGM was permitted. Participants were excluded if they had chronic kidney disease (eGFR <30ml/min); were pregnant or breastfeeding; had severe visual impairment; or had reduced manual dexterity. Participants were recruited sequentially from the young adult and general type 1 diabetes clinics at Imperial College Healthcare NHS Trust (ICHNT), UK, between 20 November 2019 and 22 July 2021. Recruitment was challenged by the COVID pandemic, but in general, participants were keen to take part. ICHNT serves an ethnically diverse population in West London, and there were no specific inclusion/exclusion criteria on ethnicity in the study.

### Semi-structured interview

All participants were invited to take part in two semi-structured interviews with a Diabetes Specialist Clinical Psychologist experienced in conducting qualitative research interviews, one within two weeks of enrolment (baseline), and another after six months of rtCGM. The interview schedule was developed by the research team. All interviews were conducted by telephone.

Research questions for the interviews were: 1) How do participants experience living with diabetes? 2) What are the barriers to self-management? 3) How did participants use rtCGM? 4) How did participants feel about using rtCGM? 5) What were the obstacles and barriers to using rtCGM? 6) What, if any, differences to living with diabetes did rtCGM make?

The baseline interview focused on participants' experiences of living with type 1 diabetes and their sense of the barriers to achieving optimal glycaemia. The post rtCGM interview focused on their experiences of using rtCGM. S1 shows the semi-structured interview schedule for baseline and post-rtCGM interviews. Interviews were scheduled for up to an hour.

To ensure confidentiality, all transcripts were pseudonymized by replacing identifying details with unique participant codes. Data were stored securely on encrypted devices, and access was restricted to authorised researchers. During transcription, any potentially identifiable information was removed to maintain participant anonymity, following established ethical guidelines.

## Psychosocial questionnaires

Participants had HbA1c checks before and after intervention and completed the following psychosocial study question-naires, which are validated tools in diabetes research and were selected to provide a picture of their general mental health, as well as diabetes-related distress specifically.

Problem areas in diabetes (PAID): This measure asks participants to select on a five-point Likert scale how much each of 20 areas of diabetes are a problem for them at present, ranging from 'not a problem' to a 'serious problem'. Scores are transformed to yield a total measure score of 0–100, with scores of ≥40 suggesting elevated diabetes distress [19].

General anxiety disorder 7 (GAD-7): A seven-item measure of anxiety, where respondents are asked the frequency with which they have experienced certain symptoms within the past two weeks ranging from 'not at all' to 'nearly every day'. The measure is scored 0–21, with a cut-off of ≥10 for caseness for anxiety [20].

Patient health questionnaire 9 (PHQ-9): A nine-item measure of depression, where respondents are asked the fre-quency with which they have experienced certain symptoms within the past two weeks from 'not at all' to 'nearly every day'. The measure is scored 0–27, with a cut-off of ≥10 for caseness for depression [21].

## Analysis

Interviews were conducted by the study psychologist (VM), recorded and transcribed verbatim, with identifying details removed. Pseudonymized transcripts were analysed using thematic analysis [22], which was selected due to its flexible theoretical methodology and rigour. An inductive, data-driven approach was used, involving repeated reviewing of inter-view transcripts to ensure familiarity with the data and to note down initial ideas. Transcripts were then coded manually (on Microsoft Word), and then collated into potential themes. This involved all codes being listed together and then clus-tered into provisional thematic groups. These were repeatedly reviewed and reconsidered to ensure the best and most coherent fit. Through this ongoing analysis, themes were reviewed, defined and named.

Credibility checks [23] involved a second researcher examining three analyzed interview transcripts to provide feedback on codes (PA), and further reviewing themes and subthemes, both individually, and together as a group (VM, PA, SM).

HbA1c and glucose data (baseline and 6 months post rtCGM initiation) were analyzed; percentage time spent in range (3.9–10 mmol/L; TIR), time spent below range (<3.9mmol/L; TBR), time spent above range (>10mmol/L; TAR). Socio-economic status was assessed by the Index of Multiple Deprivation (IMD) [24], with deprivation deciles divided into quintiles.

## Results

Eight participants were interviewed (Table 1), median age was 23.0 (22.0–24.5) years, body mass index of 25.7 kg/m$^2$ (22.1–27.1) with diabetes duration of 16.0 (8.5–17.0) years. Median baseline HbA1c was 94 (88–107) mmol/mol [10.8 (10.2–12.1)% DCCT units]. All participants used multiple daily insulin injections and 63% were currently or previously

using intermittently-scanned CGM. Participants self-reported ethnicities were Asian/Asian-British n=2; Black or Black British n=1; Mixed/Other n=5. Two participants (25%) had ketosis (beta-hydroxybutyrate >1.0 mmol/L) at recruitment. Overall percentage usage of rtCGM was 32.2 (23.1–59.4)%. Tables 2 and 3 shows individual participant data. There were no participant drop-outs during the study.

All eight participants completed both interviews which lasted 21–68 minutes (baseline) and 14–25 minutes (post-srtCGM). Table 4 outlines the four themes identified, with their associated subthemes, summarising each subtheme and providing illustrative quotations. Within the manuscript, quotations indicate which participant they are associated with (e.g., "P1"), and whether it was said in their baseline ("B") or post-rtCGM interview ("P"). The first theme, "the burden of diabetes" was drawn from baseline interviews whereas the second interview centred on use of rtCGM. The final section provides a summary of participant reflections on rtCGM.

## The burden of diabetes

At baseline, all but one of the participants spoke of difficulties and challenges of living with diabetes, citing practical, psychological and social barriers to their self-management. Practical barriers included the challenges of incorporating diabetes into busy and unpredictable lives; psychological barriers included both general mental health difficulties and diabetes-specific distress and burnout; social barriers mainly related to feeling uncomfortable or embarrassed attending to their diabetes

**Table 1. Baseline demographics.**

| Demographics (n=8) | |
|---|---|
| **Age, years** | 23.0 (22.0-24.5) |
| **Gender, male:female** | 3:5 |
| **HbA1c, mmol/mol, median (IQR)** <br> **%, median (IQR)** | 94.0 (88.3–107.3) <br> 10.8 (10.2–12.1) |
| **BMI, kg/m²** | 25.7 (22.1-27.1) |
| **Deprivation decile** | 4 (3-7) |
| **Ethnicity** | |
| White | 0 (0) |
| Asian or Asian British | 2 (25) |
| Black or Black British | 1 (12.5) |
| Mixed | 3 (37.5) |
| Other/ Not recorded | 2 (25) |
| **Insulin modality** | |
| MDI | 8 (100) |
| CSII | – |
| **Glucose sensing modality** | |
| Previous use of isCGM | 2 (25) |
| Current use of isCGM | 3 (37.5) |
| Previous/ current use of rtCGM | 0 (0) |
| **Age at diagnosis, years** | 9.0 (5.5-13.0) |
| **Duration of diabetes, years** | 16.0 (8.5-17.0) |
| **Diabetes related hospital admissions in last 18 months** | 0 (0) |
| **Overall rtCGM sensor wear, %** | 32.2 (23.1–59.4)% |

Results are expressed as median (IQR)/n (%). Abbreviations: BMI, body mass index; CSII, continuous subcutaneous insulin infusion; isCGM, intermittently scanned continuous glucose monitoring; IQR, inter-quartile range; MDI, multiple daily insulin injections; rtCGM, real-time continuous glucose monitoring.

**Table 2. Individual participant demographics and glucose data (n=8).**

| Partici-pant ID | Age at diag-nosis (years) | Use of intermit-tently scanned CGM | TIR baseline (%) | TIR endpoint (%) | % sensor use baseline | % sensor use endpoint | Overall CGM usage (%) |
|---|---|---|---|---|---|---|---|
| 1 | 3 | No | 23 | 26 | 100 | 100 | 56 |
| 2 | 14 | No | 34 | 42 | 100 | 100 | 72 |
| 3 | 7 | No | 54 | 63 | 94 | 24 | 71 |
| 4 | 9 | Yes-current | 25 | 47 | 20 | 68 | 11 |
| 5 | 12 | Yes-current | 4 | 26 | 100 | 28 | 25 |
| 6 | 4 | Yes-previous | 23 | 23 | 67 | 83 | 39 |
| 7 | 6 | Yes-current | 11 | 19 | 86 | 67 | 24 |
| 8 | 18 | Yes-previous | 0 | 13 | 39 | 86 | 19 |

Sensor usage reported in the first two weeks of sensor wear (baseline) and in the last two weeks closest to the endpoint. Overall sensor usage through-out the study also reported. Abbreviations: CGM, continuous glucose monitoring; TIR, time in range

**Table 3. Individual participant questionnaire scores.**

| Participant ID | PAID baseline | PAID endpoint | GAD-7 baseline | GAD-7 endpoint | PHQ-9 baseline | PHQ-9 endpoint |
|---|---|---|---|---|---|---|
| 1 | 64 | – | 17 | – | 21 | – |
| 2 | 44 | 16 | 0 | 2 | 6 | 3 |
| 3 | 61 | 21 | 18 | 9 | 21 | 9 |
| 4 | 39 | – | 2 | – | 8 | – |
| 5 | 13 | – | 1 | – | 5 | – |
| 6 | 34 | 43 | 2 | 9 | 4 | 5 |
| 7 | 78 | 70 | 17 | – | 21 | 23 |
| 8 | 58 | 48 | 8 | 13 | 24 | 17 |

Abbreviations: PAID, problem areas in diabetes scale; GAD-7, general anxiety disorder 7; PHQ-9, patient health questionnaire

self-management in front of other people. Two participants indicated ambivalence about the impact of their diabetes, also describing during their baseline interview how their diabetes was "not too much of a problem". Participants reported relying on physical symptoms such as "*the short-term implications of feeling sick*" [P4, B] as a cue for action. Although aware of increased risk of complications from their elevated blood glucose levels, for some, this added to a sense of feeling over-whelmed by living with diabetes and increased their anxiety, and for others, there was a sense that complications were beyond their control, or so far in the future that it was difficult for that future concern to motivate current self-management.

### Interacting with rtCGM data

The accessibility of ascertaining blood glucose level via "*just a notification on your phone*" [P6, P] was greatly appreciated, allowing convenience and discretion. All participants valued the opportunity to learn and feel more proactive in self-management. This was, however, sometimes (in seven participants) accompanied by a sense of feeling overwhelmed or frustrated, particularly when it was unclear to them why blood glucose was elevated, or if they felt unable to remedy this.

### Feelings of control and trust

All participants spoke of the control and trust experienced while using rtCGM. In some cases, this related to rtCGM allaying their concerns, such as fear of hypoglycaemia, or embarrassment from monitoring blood glucose in public.

Table 4. Themes and subthemes identified during thematic analysis with example quotations.

| Theme | Subtheme | Summary of subtheme | Illustrative quotations |
|---|---|---|---|
| **The burden of diabetes** | Diabetes isn't a big deal and I just get on with it (Ps: 4, 6, 8) | • Participants did not feel that living with diabetes had much impact on their lives, or tried not to dwell on it<br>• This was sometimes at odds with how it was described at other parts of the interview | "I don't feel like it impacts me in any way. I still do what-ever I want to do" (P4, B) |
| | Diabetes is burdensome and distressing, and affects my mental health (Ps: 1, 2, 3, 5, 6, 7, 8) | • Participants described diabetes-specific distress; diabetes contributing to general mental health problems; diabetes exacerbating mental health difficulties<br>• Fear of hypoglycaemia<br>• Fear of complications | "I do break down because of it. And I have my, my really bad days." (P7, B)<br>"almost going into a coma because of a hypo, I never ever want to go through that again in my life. So I always make sure my sugar is a certain level" (P1, B) |
| | Living in the here and now with my diabetes (Ps: 3, 4, 6, 8) | • Self-management decisions were driven by participants' immediate physical feelings<br>• Participants understood their increased risk of complications, but this did not drive their self-management behaviours<br>• Some felt that their view might only change if they developed complications | "I just don't think too deep into it because I just feel like, if it [complications] happens, it happens. I don't really think about the implications, the long-term implications. I kind of think more of the short-term implications of feeling sick, than the long term" (P4, B) |
| | I avoid diabetes, or find it hard to prioritise it (Ps: 1, 2, 4, 5, 6, 7, 8) | • Some participants actively tried to avoid thinking about or attending to their diabetes<br>• Some struggled to prioritise their diabetes within their busy, unpredictable lives | Regarding monitoring: "I don't want to know the truth. I don't really, I'd rather live thinking that I've done right, than look and see that, oh, I haven't done it right and then feel crap about it because I feel useless." (P8, B)<br>Regarding injections: "I am busy, like eating on the go or you know, I try to wait till I can do it privately, but then I didn't get the chance. I have to go rush off to do work. Or, I'm in the middle of something, stuff like that." (P2, B) |
| | I find the public aspects of living with diabetes hard (Ps: 2, 3, 5, 7, 8) | • Participants described discomfort about attending to diabetes when others could see<br>• For many this was linked to not wanting to answer questions about diabetes<br>• Embarrassment about experiencing hypoglycaemia in public | Regarding insulin: "if I'm not in, like, a private area, sometimes it does limit me and then I don't want to take it in public" (P5, B) |
| **Interacting with rtCGM data** | rtCGM is convenient and means information is readily available (Ps: 2, 3, 5, 6, 7, 8) | • Just being able to look at their phones was convenient and easy | "It's good, because you just know when your bloods are rising, if they're going low, and you don't really need to do much about it. Just a notification on your phone." (P6, P) |
| | Learning more about my diabetes and being more active in my self-management with rtCGM (Ps: 1, 2, 3, 4, 5, 8) | • Participants learned more about how their glucose was impacted by different foods and activities<br>• Feeling more involved in self-management, including being more proactive, rather than reactive | "I think I have found the pattern of what, kind of, makes me spike." (P5, P) |
| | Finding rtCGM data too much sometimes (Ps: 2, 3, 5) | • Even among those who reported rtCGM to be helpful overall, participants described finding the constant information about their glucose to be too much sometimes<br>• Frustrations of learning that glucose was high, but not knowing why or what to do about it | "you burn out quite quickly, having all constant access to this information" (P3, P) |

*(Continued)*

**Table 4.** (Continued)

| Theme | Subtheme | Summary of subtheme | Illustrative quotations |
|---|---|---|---|
| **Feelings of control and trust** | rtCGM giving peace of mind (Ps: 1, 2, 5) | • Participants felt more secure and safe knowing that they were using rtCGM | *"it's made it more manageable for me, and not having to worry, you know, am I having a hypo?"* (P2, P) |
| | Feeling more motivated, positive and in control (Ps: 1, 2, 3, 4, 5, 6, 7, 8) | • Participants described being more motivated to self-manage their diabetes due to being able to see their blood glucose readings<br>• rtCGM reduced the burden of living with diabetes<br>• Improved blood glucose readings<br>• Feeling better about living with diabetes when using rtCGM, including feeling more normal and free | *"let's say I was able to see that it was going high, or going low, I would do some sort of action to counteract it. And it kind of made me feel more motivated to actually take care of myself."* (P8, P)<br>*"something like this just makes handling the condition easier. And it kind of put me in the mind frame like, I am more in control of this and you know, it's not such a big deal as people are making it out to be."* (P2, P) |
| | Alarms let me know if I need to do anything (Ps: 4, 5, 8) | • Participants used alarms a cue for action<br>• Only using alarms and rarely looking at rtCGM readings otherwise | *"I know that if I don't get the beep, then everything is fine."* (P5, P) |
| | rtCGM is more discrete (Ps: 1, 2, 5, 7) | • Participants could monitor their blood glucose without others knowing what they were doing<br>• The sensor was more discrete than isCGM | Regarding isCGM: *"friends would ask me, what is that? Like, I would just hate to, to explain the whole thing all over again and it's just frustrating. So, erm, but with this one, you just put it on the stomach, so it was hidden most of the time."* (P7, P) |
| **Frustration of tech and alerts/ alarms** | Not always appreciating alarms (Ps: 3, 5, 6) | • Alarms could be noisy and embarrassing<br>• Alarms woke participants which disrupted sleep<br>• Alarms could sound for a time after participants had treated their blood glucose, which was frustrating<br>• Disabling alarms and wanting to have more control over which alarms could be disabled | *"Just alarms, yeah just, it would ring like while I was in class."* (P6, P)<br>*"I don't need the constant alarms. I just need it when it's low, because that's what, something that would require immediate action."* (P3, P) |
| | Finding the physical and technical aspects of rtCGM difficult and frustrating (Ps: 4, 5, 6, 7, 8) | • Participants experienced challenges with the rtCGM technology, especially using the app<br>• Finding rtCGM too high-tech<br>• Finding the sensor fiddly, and having difficulties with successfully inserting sensors | *"I had to keep deleting the app and putting it back on. And every time if I didn't scan my blood sugars for a couple of hours, it would do the same thing, so I found that quite fiddly"* (P4, P) |

The first theme reflects the burden of self-management of diabetes, with the other three themes related to the use of rtCGM. Quotations indicate which participant they are associated with, and whether it was said in their baseline ("B") or post-rtCGM interview ("P"). Abbreviations: Ps, participants; rtCGM, real-time continuous glucose monitoring.

rtCGM enabled some participants to continue with their preferred self-management style, such as being focused on the here-and-now, using alarms as a cue for action: "*I know that if I don't get the beep, then everything is fine*" [P5, P]. All participants described increased motivation for, or engagement in, their self-management, and/or improved feelings about themselves or living with diabetes with rtCGM: "*it kind of made me feel more motivated to actually take care of myself*" [P8, P].

### Frustration of technology and alarms

The experience of alarms was mixed, within and between participants. Alarms were described as a source of frustration by three participants, for example waking them at night, causing embarrassment in public places, or continuing to sound even after taking corrective action. Participants also described challenges interacting with rtCGM technology, in particular using the phone application and having to reinstall on repeated occasions. For three participants, these frustrations resulted in them taking a break or completely stopping use before study completion. The five participants who reported technical difficulties with rtCGM (Participants 4, 5, 6, 7, 8) had the lowest sensor usages (all under 40% wear time), and some reported that they would have worn it more had those challenges been resolved.

### Overall participant reflections on rtCGM

Participants often reported feeling that the opportunity to use rtCGM for the present study was likely to be "*kind of, a waste of time*" [P2, P], or that "*when I was first told about it…I wasn't really sure*" [P1, P]. Despite this, and despite relatively low levels of rtCGM wear time, all participants reported some level of improved diabetes self-management and/or feelings about themselves or about living with diabetes while using rtCGM.

Six participants said that they would wish to continue using rtCGM if available. Participant 4 reported finding the rtCGM useful, but too "*fiddly*" [P4, P] and said that they "*don't really want to commit any more time towards diabetes*". Participant 6 reported liking rtCGM and finding it useful, saying that they had a better quality of life while using rtCGM, apart from low glucose alarms which woke them at night, and finding the device "*too high-tech*" [P6, P]. Due to the disruption from these alarms, Participant 6 said that they would not wish to continue to use rtCGM.

## Discussion

Findings from this study in young adults from minoritised ethnicities with very elevated HbA1c show a complex relationship in use and engagement with rtCGM. Thematic analysis of participant interviews indicated that participants made use of rtCGM in multiple ways, some addressing self-management barriers, such as avoiding glucose monitoring in public, and some assisting their preferred style of being focused on the present moment, relying on alarms to indicate necessary action. Although characterised by high levels of diabetes distress from finding diabetes burdensome, participants did not universally find rtCGM beneficial. This was identified across three themes during interviews: 1) interaction with data: conflicting feelings around data availability leading to better engagement and knowledge versus feeling overwhelmed by the constant stream of data; 2) control and trust: the majority felt they had better control and felt more motivated and 3) frustration: reports from many regarding technical difficulties and disruptive alerts.

Sensor wear-time was low but despite this, some metrics of glycaemia showed improvement. It has been established that rtCGM use is associated with a significant reduction in HbA1c, which is greatest in those with the highest Hba1c at baseline and who more frequently used the sensors (more than 70% of the time or near continuously) [12]. Thus, supporting individuals to address sensor usage, is likely to further improve glycaemia. For those with lower HbA1c groups, CGM can reduce exposure to hypoglycaemia [8,12]. We noted levels of diabetes distress and depression were high. Although rtCGM may have potential to improve glycaemic outcomes in very high-risk young adults, the high levels of diabetes distress and depression likely contributed to making interactions with the technology burdensome to most.

Participants' experience of alarms was mixed, and in keeping with previous research [25], the perception of alarms as helpful or frustrating typically depended on perceived utility of the alarm at that time. Alarm fatigue has been described in diabetes devices, especially in CGM systems. Device burden is known to be associated with discontinuation of CGM use [26], as was demonstrated in our study.

General "tech savviness" can play a role in the adoption of CGM technologies [27]. Young adults, who may be assumed to be confident and competent in using technology, reported challenges with interacting with rtCGM technology as a key reason for reduced use or disengagement from rtCGM. This highlights the need for further practical rtCGM support and education as part of regular clinical care for this group. Many participants described being more engaged in their diabetes self-management while using rtCGM, but one participant's consequent experience of being repeatedly woken at night by hypoglycaemia alarms, and therefore stopping rtCGM early, further illustrates the importance of ongoing support and education around integrating rtCGM-provided information into one's diabetes self-management [27]. Using rtCGM data for retrospective pattern analysis is associated with improved glycaemic outcomes compared to minute-by-minute data analysis [28], which may be a particular area of education to consider for this cohort.

Some individuals, particularly those from marginalised groups may not be aware of the availability or have previously been offered rtCGM [29]. However, it is likely that with appropriate support and education, this group can attain some benefits.

The use of rtCGM demands extra effort on the part of the young person with type 1 diabetes, therefore it is important to assess if this extra burden leads to psychosocial distress [26]. Amongst adolescents and young adults, international registries have consistently reported higher HbA1c levels compared with pre-adolescents and older adults with type 1 diabetes [6,7]. There are multiple challenges experienced by this age group, including physical issues, financial concerns, emotional burdens, and device inadequacies [30,31]. Such barriers may extend to prevent useful engagement with technology.

## Clinical impact of this study

The unique contributions of this study relate to the evaluation of psychosocial factors associated with rtCGM use, within a high-risk cohort of individuals, from ethnicities typically under-represented in research. Despite the availability of new technologies for diabetes management, suboptimal glycaemic control remains common among some young adults with type 1 diabetes.

Our results show that even with relatively low sensor wear time there may be glycaemic benefits yielded from rtCGM in young adults with extreme hyperglycaemia with vulnerable characteristics. However, it is important that this benefit is not outweighed by technology contributing to diabetes distress and burnout. Our results indicate that more support around alarm setting (such as allowing judicious choice of thresholds to minimise nuisance alarms and retain only important ones), technology set up (for example support with installing and interacting with the mobile phone application) and troubleshooting (for example being able to contact a professional for technical support), may help some individuals to avoid feeling overwhelmed. This may in turn lead to greater benefits yielded from rtCGM.

A key finding of this study was that participants reported practical and technical challenges with using rtCGM, which affected their sensor wear time and general engagement. We recommend this cohort to benefit from additional support in using such technologies with more intensive training and psychosocial support. As there is a shift towards more virtual rtCGM initiations or indeed, self-starts using online educational videos, it is important to consider that younger adults may need more intensive support when starting rtCGM than may be assumed.

## Strengths and limitations

The strengths of this study include selecting individuals at highest risk with extreme hyperglycaemia, from ethnicities which tend to be underrepresented in type 1 diabetes research. In the present study, we targeted individuals with a higher HbA1c (> 75 mmol/mol) and reported a median HbA1c of 94mmol/mol, compared to the MILLENIALS study, where the

mean HbA1c at baseline was 78.4mmol/mol. It is likely the present study captured more people with extreme hypergly-caemia; indeed two participants were ketotic at screening, indicating high rates of insulin omission.

In terms of limitations, baseline time in range was assessed using rtCGM in the first weeks of usage, rather than blinded intervention. Due to the change in national guidance supporting access to glucose monitoring technology for all people with type 1 diabetes, the study was terminated early. This means that only eight participants were recruited, and both of these issues may have limited the findings. Although by later interviews, we had started to reach the judgement that data saturation [32] had been achieved, it would have been ideal to recruit several more participants in order to ensure this. We anticipate that readers will be able to ascertain which aspects of this study's findings would be transferra-ble within different contexts [33].

Sensor wear time was low. It is not clear if this was due to technical issues or other reasons. Future studies that provide individuals from similar contexts with bespoke and intensive support around rtCGM would help to ascertain if wear time increases and HbA1c improves with such support, as well as provide more information about the psychosocial benefits and possible challenges that may arise when that support package is in place. Further studies with real-world experience on recognition and understanding psychological barriers are required to support young adults using technology.

## Conclusions

Our qualitative analysis demonstrates that vulnerable young adults with extreme hyperglycaemia have a complex rela-tionship with rtCGM whereby its positive impact in supporting self-management is often offset by effort and feeling over-whelmed by data and alarms. The significant psychological burden that remains with rtCGM use and the requirement for ongoing support and education needs to be addressed when considering initiation of rtCGM in people with high-risk hyperglycaemia and/or those experiencing significant burden from self-management. Further research is required on how to engage and support young adults in the highest risk groups (i.e., HbA1c > 75 mmol/mol (9%) or ≥ 1 DKA admissions or ≥ 1 hospital admission with extreme hyperglycaemia) with rtCGM, and whether a more structured educational and psy-chosocial support package with greater contact time than is the current standard of care, and specific technical support, may offset challenges from technology usage. These research findings will influence clinical practice through highlighting the need for better integration of rtCGM into young adult T1D management through alternative care models, such as more frequent check-ins and targeted interventions for high-risk groups.

## Supporting information

**S1 File. Semi-structured interview schedule for baseline and post-rtCGM interviews.**
(DOCX)

## Author contributions

**Conceptualization:** Vicky McKechnie, Rebecca Unsworth, Monika Reddy, Shivani Misra.

**Data curation:** Vicky McKechnie, Parizad Avari, Pei Chia Eng, Rebecca Unsworth, Shivani Misra.

**Formal analysis:** Vicky McKechnie, Parizad Avari.

**Investigation:** Vicky McKechnie, Parizad Avari, Pei Chia Eng, Rebecca Unsworth.

**Methodology:** Vicky McKechnie, Rebecca Unsworth, Shivani Misra.

**Project administration:** Parizad Avari, Pei Chia Eng, Rebecca Unsworth.

**Supervision:** Shivani Misra.

**Writing – original draft:** Vicky McKechnie, Parizad Avari.

**Writing – review & editing:** Parizad Avari, Monika Reddy, Stephanie A. Amiel, Victoria Salem, Shivani Misra.

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
