## [Decision Letter · Decision Letter 0]

11 Nov 2024

PONE-D-24-45438The experiences of high-risk young adults with type 1 diabetes transitioning to real-time continuous glucose monitoring – A thematic analysisPLOS ONE

Dear Dr. Misra,

Thank you for submitting your manuscript to PLOS ONE. After careful consideration, we feel that it has merit but does not fully meet PLOS ONE’s publication criteria as it currently stands. Therefore, we invite you to submit a revised version of the manuscript that addresses the points raised during the review process.

We look forward to receiving your revised manuscript.

Kind regards,

Yusuf Oloruntoyin Ayipo, Ph.D

Academic Editor

PLOS ONE

Journal Requirements:

“This study was funded by an investigator-initiated grant from DexCom Inc.”

“SM is funded by a Wellcome Trust Career Development Award (223024/Z/21/Z) and is supported by the NIHR Imperial Biomedical Research Centre. PA is funded by NIHR Grant No. 132960. VS is a Diabetes UK Grand Challenge Senior Fellow as is also supported by grants currently held from the MRC, BBSRC and NC3Rs. Infrastructure support was provided by the NIHR Imperial Biomedical Research Centre and the NIHR Imperial Clinical Research Facility. PCE is supported by the NUS Exxon Mobil Grant and the National Medical Research Council (NMRC) New Investigator Grant, Singapore.”

“This study was funded by an investigator-initiated grant from DexCom Inc.”

“I have read the journal's policy and the authors of this manuscript have the following competing interests: SM serves as Trustee to the Diabetes Research & Wellness Foundation charity, UK and has received speaker Honoraria from Lilly and Sanofi, UK. MR has received honoraria for advisory board participation from Dexcom and Roche Diabetes. SAA has served on Advisory Boards for Novo Nordisk and Medtronic and has spoken at educational meetings supported by NovoNordisk and Sanofi. VM, PA, PCE, RU have no conflicts of interest to declare.”

5. We note that you have indicated that there are restrictions to data sharing for this study. For studies involving human research participant data or other sensitive data, we encourage authors to share de-identified or anonymized data. However, when data cannot be publicly shared for ethical reasons, we allow authors to make their data sets available upon request. For information on unacceptable data access restrictions, please see http://journals.plos.org/plosone/s/data-availability#loc-unacceptable-data-access-restrictions. 

6. Please amend the manuscript submission data (via Edit Submission) to include author “DClinPsy”.

**Additional Editor Comments:**

The reviewers have raised some concerns that require authors’ attention significantly before the manuscript can be recommended for publication in this journal.

Reviewers' comments:

Reviewer's Responses to Questions

**Comments to the Author**

1. Is the manuscript technically sound, and do the data support the conclusions?

Reviewer #1: Partly

Reviewer #2: Yes

Reviewer #3: Yes

Reviewer #4: Yes

2. Has the statistical analysis been performed appropriately and rigorously? 

Reviewer #1: N/A

Reviewer #2: N/A

Reviewer #3: Yes

Reviewer #4: Yes

3. Have the authors made all data underlying the findings in their manuscript fully available?

Reviewer #1: Yes

Reviewer #2: Yes

Reviewer #3: Yes

Reviewer #4: Yes

4. Is the manuscript presented in an intelligible fashion and written in standard English?

Reviewer #1: Yes

Reviewer #2: Yes

Reviewer #3: Yes

Reviewer #4: Yes

5. Review Comments to the Author

Reviewer #1: This manuscript examines the psychosocial impact of rtCGM in high-risk young adults with type 1 diabetes, focusing on individuals with very high HbA1c levels who are typically excluded from such studies. The study addresses a crucial area and contributes valuable insights, but several areas need refinement to strengthen the impact of the manuscript.

1. In the introduction, the authors mention that individuals with high HbA1c are at greater risk for emergencies and insulin omission. They should further explain why these characteristics are particularly relevant to studying the psychosocial impact of rtCGM, clarifying why this group is uniquely suited for this research.

2. Small grammar error in (line 42-43) "the psychosocial impact of rtCGM on in young adults".

3. The authors note that previous rtCGM studies have excluded individuals with high HbA1c due to safety concerns, yet they do not address how these concerns are managed within their study. If this study includes individuals with very high HbA1c despite known risks, the authors should briefly describe any precautions or study adaptations that were made to ensure participant safety, which would justify why the benefits of inclusion outweigh these potential risks.

4. Terms like “very high-risk” and “suboptimal HbA1c” (lines 108-109) are vague. Define the specific thresholds used in this study and explain why these values represent “very high risk” in diabetes care, with reference to clinical standards or existing literature.

5. The claim of “diverse ethnic backgrounds” (line 391) seems inconsistent with the demographic data provided, as the sample has only 8 individuals, is homogeneously non-white and only represents a limited range of ethnic groups. Rather than labeling the sample as “diverse,” it would be more accurate to emphasize the value of including underrepresented backgrounds in diabetes research. Additionally, the limitations of this demographic focus on generalizability should be acknowledged.

6. The thematic analysis process lacks sufficient detail. Specify the coding approach (NVivo, manual coding) and outline each step in the process, including how themes were refined. This is essential for reproducibility.

7. Address data saturation in more detail, particularly given the small sample size. Explain how saturation was determined with only eight participants and whether the early termination of the study might have prevented full thematic saturation.

8. Sensor wear-time was notably low, which is an interesting finding that requires further analysis. Identify and discuss specific “practical and technical challenges” that contributed to low adherence based on participant feedback.

I suggest publication after addressing these comments.

Reviewer #2: Strengths:

Addresses an important gap in literature regarding high-risk young adults with extreme hyperglycemia

Unique focus on ethnically diverse population often underrepresented in research

Mixed-methods approach combining qualitative data with clinical metrics

Clear presentation of themes with supporting quotes

Balanced discussion of both benefits and challenges of rtCGM

Strong clinical implications identified

Minor Suggestions for Improvement:

Consider adding a flow diagram showing participant recruitment and retention

Provide more detail about the credibility checking process

Include information about interview duration ranges in methods

Consider reorganizing Table 2 for better readability

Add brief explanation of why specific psychosocial questionnaires were chosen

Reviewer #3: The manuscript is generally well organized with good flow of ideas from the introduction to the conclusions. The methodology is mostly well described. Overall, the data presentation and interpretation are clear as conclusions fully support the data presented. The result section is comprehensive, and the discussion section is well utilized and beneficial as you compared your findings with similar results, and that was able to strengthen your argument and showed the relevance of your study. Figures and tables are informative and effectively illustrated your results. The manuscript is generally well written, however, the reference style format do not follow the Vancouver style format. I recommend you visit the journal website to see a sample of the Vancouver style format.

Reviewer #4: 1. Expand on the recruitment process and selection criteria for participants, particularly how the specific demographic (non-white, high-risk young adults) was chosen. This will help readers understand the representativeness of the sample.

2. Given the low wear-time of the rtCGM among participants, consider discussing how this limitation might have impacted study outcomes, particularly the generalizability of findings regarding the psychosocial benefits of rtCGM.

3. Provide more detail on the thematic analysis, including steps taken for coding and theme identification. Consider adding credibility checks or inter-rater reliability measures to strengthen the qualitative analysis.

4. Since alarm fatigue was a significant barrier to rtCGM adherence, delve deeper into possible solutions, such as customizing alarm settings or offering educational support to minimize alarm-related distress.

5. Provide a more comprehensive review of existing literature on CGM usage in populations with high HbA1c levels, noting any contrasts or similarities to findings in lower HbA1c groups.

6. Given the technical challenges reported, recommend integrating more intensive training or technical support for users, particularly young adults who might be less familiar with CGM technology.

7. Since high diabetes distress and burnout were noted, discuss in greater depth the need for psychosocial support interventions as part of rtCGM adoption, particularly for high-risk groups.

8. Acknowledge how the small sample size and early study termination due to policy changes might have limited the findings, and suggest how future research can address these constraints for more robust data.

6. PLOS authors have the option to publish the peer review history of their article (what does this mean? ). If published, this will include your full peer review and any attached files.

**Do you want your identity to be public for this peer review?** For information about this choice, including consent withdrawal, please see our Privacy Policy .

Reviewer #1: No

Reviewer #2: **Yes: ** Kelechi Wisdom Elechi

Reviewer #3: No

Reviewer #4: **Yes: ** Adewunmi Akingbola

---

## [Author Response · Author response to Decision Letter 1]

24 Jan 2025

Dear Editor,

Thank you very much for your helpful feedback on ways in which we can improve our manuscript entitled: The experiences of high-risk young adults with type 1 diabetes transitioning to real-time continuous glucose monitoring – A thematic analysis (PONE-D-24-45438). Please find below our responses to the comments highlighted in bold. The new changes in the manuscript are highlighted in blue.

Regarding the journal requirements:

1. We have reviewed the manuscript and made a small number of changes to ensure that it meets the journal’s style requirements.

2. We have removed funding information from the manuscript (funding statement and acknowledgements section). Thank you for amending the online submission form to indicate the following financial disclosure:

“Dexcom provided funding for sensors and research staff time. The funder of the study had no role in study design, data collection, data analysis, data interpretation, or writing of the report.

SM is funded by a Wellcome Trust Career Development Award (223024/Z/21/Z) and is supported by the NIHR Imperial Biomedical Research Centre. PA is funded by NIHR Grant No. 132960. VS is a Diabetes UK Grand Challenge Senior Fellow as is also supported by grants currently held from the MRC, BBSRC and NC3Rs. PCE is supported by the NUS Exxon Mobil Grant and the National Medical Research Council (NMRC) New Investigator Grant, Singapore.”

3. Please see above.

4. I confirm that none of the authors’ competing interests alter our adherence to PLOS ONE policies on sharing data and materials.

5. In line with PLOS ONE guidelines for qualitative data, we have shared excerpts of transcripts within the manuscript. Sharing larger amounts of the qualitative data would risk participant identification. However, reasonable requests for further data can be sent to the corresponding author for consideration.

6. We have removed author qualifications from the list of authors in the manuscript.

Thank you for the helpful reviewer comments, which we address below.

Reviewer 1:

This manuscript examines the psychosocial impact of rtCGM in high-risk young adults with type 1 diabetes, focusing on individuals with very high HbA1c levels who are typically excluded from such studies. The study addresses a crucial area and contributes valuable insights, but several areas need refinement to strengthen the impact of the manuscript.

1. In the introduction, the authors mention that individuals with high HbA1c are at greater risk for emergencies and insulin omission. They should further explain why these characteristics are particularly relevant to studying the psychosocial impact of rtCGM, clarifying why this group is uniquely suited for this research.

Thank you for this comment. We have added a sentence in the introduction to clarify this. The relevant paragraph now reads as follows (new text italicised):

“Many rtCGM studies have excluded people with extremely high HbA1c due to safety concerns; the mean HbA1c (14-25-year-olds) in the JDRF-CGM study was 63.9mmol/mol (8%) (1) and 78.4mmol/mol (9.3%) in the MILLENNIALS study(2). Individuals with much higher HbA1c are likely to be those at highest risk of hyperglycaemic emergencies and insulin omission and are also individuals who may struggle with self-management due to other factors such as mental well-being or burnout.”

2. Small grammar error in (line 42-43) "the psychosocial impact of rtCGM on in young adults".

Thank you for pointing this out. We have corrected it.

3. The authors note that previous rtCGM studies have excluded individuals with high HbA1c due to safety concerns, yet they do not address how these concerns are managed within their study. If this study includes individuals with very high HbA1c despite known risks, the authors should briefly describe any precautions or study adaptations that were made to ensure participant safety, which would justify why the benefits of inclusion outweigh these potential risks.

The study was designed to assess the impact of CGM and the experience of high-risk young adults with type 1 diabetes using rtCGM. Apart from the study visits and procedures, no specific study adaptations were made, to reflect real-world acceptability and experiences. The provision of CGM is now standard care in the National Health Service in England, for all individuals with type 1 diabetes, therefore the provision of CGM itself was not deemed to be an additional risk over and above the high risk posed by their self-management practices.

4. Terms like “very high-risk” and “suboptimal HbA1c” (lines 108-109) are vague. Define the specific thresholds used in this study and explain why these values represent “very high risk” in diabetes care, with reference to clinical standards or existing literature.

The specific thresholds used in the study are included in the methods (Page 7, Paragraph on Participants) and include “one or more of: HbA1c > 75 mmol/mol (9%) or ≥ 1 DKA admissions or ≥ 1 hospital admission with extreme hyperglycaemia in the last 18 months.”

We have added the following explanation for why these represent a very high risk: “These criteria were based on people with recurrent DKA having higher HbA1c(3) and readmission rates(4). Many early rtCGM studies have excluded people with extremely high HbA1c due risks of DKA.”

5. The claim of “diverse ethnic backgrounds” (line 391) seems inconsistent with the demographic data provided, as the sample has only 8 individuals, is homogeneously non-white and only represents a limited range of ethnic groups. Rather than labeling the sample as “diverse,” it would be more accurate to emphasize the value of including underrepresented backgrounds in diabetes research. Additionally, the limitations of this demographic focus on generalizability should be acknowledged.

Thank you for this point. We note two references to “diverse ethnicities” in the manuscript and have adjusted these, as suggested, to read “from ethnicities typically under-represented in research”.

We note that qualitative research, often conducted within the framework of a constructivist epistemology, would not aim to be generalisable, but rather would aim to offer sufficient information about the participants, procedure and results for consumers of the researchers to determine which aspects might be transferrable in which contexts. To address the reviewer’s comments, we have included in the strengths and limitations section of the discussion a reflection about transferability, which reads: “We anticipate that readers will be able to ascertain which aspects of this study’s findings would be transferrable within different contexts.”

6. The thematic analysis process lacks sufficient detail. Specify the coding approach (NVivo, manual coding) and outline each step in the process, including how themes were refined. This is essential for reproducibility.

Thank you. We have made some additions to the information about the thematic analysis process which can be found on page 9 of the manuscript with tracked changes. It now reads: “Interviews were conducted by the study psychologist (VM), recorded and transcribed verbatim, with identifying details removed. Pseudonymized transcripts were analysed using thematic analysis (5). An inductive, data-driven approach was used, involving repeated reviewing of interview transcripts to ensure familiarity with the data and to note down initial ideas. Transcripts were then coded manually (on Microsoft Word), and then collated into potential themes. This involved all codes being listed together and then clustered into provisional thematic groups. These were repeatedly reviewed and reconsidered to ensure the best and most coherent fit. Through this ongoing analysis, themes were reviewed, defined and named.”

7. Address data saturation in more detail, particularly given the small sample size. Explain how saturation was determined with only eight participants and whether the early termination of the study might have prevented full thematic saturation.

Thank you for this important point; as included in the methods, the study was terminated early for practical reasons “due to new NICE guidance published in 2022 advocating rtCGM for all adults with type 1 diabetes as standard care”. We acknowledge in the discussion “it would have been ideal to recruit several more participants”. However, the thematic analyses was conducted by an experienced highly specialist diabetes psychologist and we have included the line “by later interviews, we had started to reach the judgement that data saturation had been achieved”.

8. Sensor wear-time was notably low, which is an interesting finding that requires further analysis. Identify and discuss specific “practical and technical challenges” that contributed to low adherence based on participant feedback.

On page 18 of the results, we discuss the specific practical and technical challenges that participants reported were having an impact on their interaction with the rtCGM technology. The themes identified 1) interaction with rtCGM data, 2) feelings of control and trust from using rtCGM, and 3) frustration of technology and alarms, are all likely to contribute to low sensor adherence.

In the discussion, we consider possible ways to improve these aspects and support individuals. We have added some more specific recommendations on page 22 to support young adults with rtCGM. “Our results indicate that more support around alarm setting (such as allowing judicious choice of thresholds to minimise nuisance alarms and retain only important ones), technology set up (for example support with installing and interacting with the mobile phone application) and troubleshooting (for example being able to contact a professional for technical support), may help some individuals to avoid feeling overwhelmed.”

Reviewer 2:

Strengths:

Addresses an important gap in literature regarding high-risk young adults with extreme hyperglycaemia. Unique focus on ethnically diverse population often underrepresented in research. Mixed-methods approach combining qualitative data with clinical metrics

Clear presentation of themes with supporting quotes. Balanced discussion of both benefits and challenges of rtCGM. Strong clinical implications identified

Minor Suggestions for Improvement:

Consider adding a flow diagram showing participant recruitment and retention

Thank you. We have clarified there were no participant drop-outs in the study, and all 8 participants completed both interviews.

Provide more detail about the credibility checking process

We have added more information about this on page 10, which now reads (new text in italics): “Credibility checks (6) involved a second researcher examining three analyzed interview transcripts to provide feedback on codes (PA), and further reviewing themes and subthemes, both individually, and together as a group (VM, PA, SM).”

Include information about interview duration ranges in methods

Thank you – we have added the interview duration in the methods. “Interviews were scheduled for up to an hour”

We have also outlined the duration of each interview in the Results section on page 18.

Consider reorganizing Table 2 for better readability

Thank you. We have separated it into two tables (Tables 2 and 3) and hope this is easier to read.

Add brief explanation of why specific psychosocial questionnaires were chosen

We have updated page 8, and it now reads (new part in italics): “Participants had HbA1c checks before and after intervention and completed the following psychosocial study questionnaires, which are validated tools in diabetes research and were selected to provide a picture of their general mental health, as well as diabetes-related distress specifically.”

Reviewer 3:

The manuscript is generally well organized with good flow of ideas from the introduction to the conclusions. The methodology is mostly well described. Overall, the data presentation and interpretation are clear as conclusions fully support the data presented. The result section is comprehensive, and the discussion section is well utilized and beneficial as you compared your findings with similar results, and that was able to strengthen your argument and showed the relevance of your study. Figures and tables are informative and effectively illustrated your results. The manuscript is generally well written, however, the reference style format do not follow the Vancouver style format. I recommend you visit the journal website to see a sample of the Vancouver style format.

Thank you for the positive comments. We have updated the manuscript references to Vancouver format.

Reviewer 4:

1. Expand on the recruitment process and selection criteria for participants, particularly how the specific demographic (non-white, high-risk young adults) was chosen. This will help readers understand the representativeness of the sample.

There were no specific criteria on demographics (i.e. non-white participants). This is likely to have reflected the ethnically diverse diabetes service from which participants were recruited in London. We have included an additional sentence for clarification in the methods: “Imperial College Healthcare NHS Trust (ICHNT) serves an ethnically diverse population in West London, and there were no specific inclusion/exclusion criteria on ethnicity in the study.”

2. Given the low wear-time of the rtCGM among participants, consider discussing how this limitation might have impacted study outcomes, particularly the generalizability of findings regarding the psychosocial benefits of rtCGM.

Thank you for this helpful comment. We have added a reflection in the discussion that reads: “Sensor wear time was low. It is not clear if this was due to technical issues or other reasons. Future studies that provide individuals from similar contexts with bespoke and intensive support around rtCGM would help to ascertain if wear time increases and HbA1c improves with such support, as well as provide more information about the psychosocial benefits and possible challenges that may arise when that support package is in place.”

3. Provide more detail on the thematic analysis, including steps taken for coding and theme identification. Consider adding credibility checks or inter-rater reliability measures to strengthen the qualitative analysis.

Thank you. Reviewer 1 also asked for more information about the thematic analysis process, and this has now been added on page 9 and includes a description of credibility checks.

4. Since alarm fatigue was a significant barrier to rtCGM adherence, delve deeper into possible solutions, such as customizing alarm settings or offering educational support to minimize alarm-related distress.

Thank you – We have expanded on the comments in the discussion about customising alarm thresholds (Page 22). With regards to minimising distress: “This cohort may benefit from additional support in using such technologies with more intensive training and psychosocial support”. The final sentence of the manuscript has also been updated to: “Further research is required on how to engage and support young adults in the highest risk groups, and whether a more structured educational and psychosocial support package with greater contact time than standard care, and specific technical support, may offset challenges from technology usage.”

5. Provide a more comprehensive review of existing literature on CGM usage in populations with high HbA1c levels, noting any contrasts or similarities to findings in lower HbA1c groups.

Thank you for raising this important point. The association of CGM sensor usage has been well established in populations with high HbA1c levels. We have included the following statement: “It has been established that rtCGM use is associated with a significant reduction in HbA1c, which is greatest in those with the highest Hba1c at baseline and who more frequently used the sensors (more than 70% of the time or near continuously) (7,8). Thus, supporting individuals to address sensor usage, is likely to further improve glycaemia. For those with lower HbA1c groups, CGM can reduce exposure to hypoglycaemia(8,9).”

We have also included in the following paragraph on pag

---

## [Decision Letter · Decision Letter 1]

9 Feb 2025

PONE-D-24-45438R1The experiences of high-risk young adults with type 1 diabetes transitioning to real-time continuous glucose monitoring – A thematic analysisPLOS ONE

Dear Dr. Misra,

Thank you for submitting your manuscript to PLOS ONE. After careful consideration, we feel that it has merit but does not fully meet PLOS ONE’s publication criteria as it currently stands. Therefore, we invite you to submit a revised version of the manuscript that addresses the points raised during the review process.

**ACADEMIC EDITOR: **The authors have responded positively to the initial critiques. However, some minor revisions are still required to support the quality of the submission. I invite the authors to revise according to the comments of Reviewer 6. The abstract should be structured according to the journal's guidelines.

We look forward to receiving your revised manuscript.

Kind regards,

Yusuf Oloruntoyin Ayipo, Ph.D

Academic Editor

PLOS ONE

Journal Requirements:

Additional Editor Comments:

The authors have responded positively to the initial critiques. However, some minor revisions are still required to support the quality of the submission. I invite the authors to revise according to the comments of Reviewer 6. The abstract should be structured according to the journal's guidelines.

Reviewers' comments:

Reviewer's Responses to Questions

**Comments to the Author**

1. If the authors have adequately addressed your comments raised in a previous round of review and you feel that this manuscript is now acceptable for publication, you may indicate that here to bypass the “Comments to the Author” section, enter your conflict of interest statement in the “Confidential to Editor” section, and submit your "Accept" recommendation.

Reviewer #1: All comments have been addressed

Reviewer #4: All comments have been addressed

Reviewer #5: All comments have been addressed

Reviewer #6: (No Response)

2. Is the manuscript technically sound, and do the data support the conclusions?

Reviewer #1: Yes

Reviewer #4: Yes

Reviewer #5: Yes

Reviewer #6: Yes

3. Has the statistical analysis been performed appropriately and rigorously? 

Reviewer #1: N/A

Reviewer #4: Yes

Reviewer #5: Yes

Reviewer #6: N/A

4. Have the authors made all data underlying the findings in their manuscript fully available?

Reviewer #1: Yes

Reviewer #4: Yes

Reviewer #5: Yes

Reviewer #6: Yes

5. Is the manuscript presented in an intelligible fashion and written in standard English?

Reviewer #1: Yes

Reviewer #4: Yes

Reviewer #5: Yes

Reviewer #6: Yes

6. Review Comments to the Author

Reviewer #1: In my opinion the authors have adequately addressed the reviewers comments and the manuscript is fit for publication.

Reviewer #4: A good job to the authors of this manuscript for their hardworking. I wish you all the best in your scientific journey.

Reviewer #5: This study explores the psychosocial impact of real-time continuous glucose monitoring (rtCGM) on high-risk young adults with type 1 diabetes, a demographic that has not been extensively studied. The authors conducted a qualitative investigation involving eight participants aged 18–25 years with poorly controlled diabetes (HbA1c >75mmol/mol) who were naïve to rtCGM. Participants used rtCGM for six months, and semi-structured interviews were conducted at recruitment and study completion to assess barriers to self-management and experiences with rtCGM. Thematic analysis revealed three key themes: 1) interaction with rtCGM data, 2) feelings of control and trust, and 3) frustration with technology and alarms. Despite low wear-time (32.2%), improvements in time in range were observed, though HbA1c levels remained unchanged. Participants reported convenience and greater control from accessing glucose data on their smartphones, but these benefits were offset by alarm fatigue, technical difficulties, and feelings of being overwhelmed. Three participants discontinued rtCGM prematurely, highlighting the complex relationship high-risk young adults have with this technology. The authors conclude that while rtCGM may offer benefits, its use in this population requires additional support and individualized consideration to mitigate potential distress or burnout.

The manuscript is technically sound, with data supporting the conclusions. The research was conducted rigorously, with appropriate controls and sample sizes. Although the sample size for the study was small, the authors justified it in the limitation of the study. Statistical analysis was performed appropriately, and the authors have made all underlying data fully available. The manuscript is well-written, intelligible, and adheres to standard English. The authors have adequately addressed previous review concerns, and the manuscript is hereby recommended to be accepted for publication.

Reviewer #6: PEER REVIEW REPORT

Abstract Section

1. The abstract successfully introduces the relevance of rtCGM for type 1 diabetes care in a particular population but does not provide a defined explanation of the knowledge gap it intends to address regarding psychosocial effects.

2. The abstracts methods section summarizes participant demographics and data collection and analysis methods but does not provide specific information about the study design . Kindly state the study design e.g., "A qualitative study employing semi-structured interviews...").

3. The results section: In addition to the primary findings and thematic insights outlined, consider including a brief quantifications of the major outcomes

4. Consider using structured abstracts formatted with headings such as Background, Objectives, in addition to Methods, Results, and Conclusions already included.

5. Review the conclusion to ensure it connects the study findings with its objectives and the research gap more effectively.

Introduction Section

1. The introduction establishes a solid foundation through its discussion of intensive self-management for type 1 diabetes and the challenges faced by young adults. Consider providing explicit definition of the research gap concerning the psychosocial effects of rtCGM for high-risk group.

2. The introduction could be strengthened by explicitly specifying the research questions or hypotheses.

Method Section

1. Participant sub-section: Provide more comprehensive detail of the recruitment process. Briefly discuss challenges encountered during recruitment such as participants willingness and recruitment rate.

2. Although you mentioned that participant were recruited from Imperial College Healthcare NHS Trust diabetes clinics in the UK . Kindly provide information on how the clinic was selected,.

3. Analysis sub-section: Provide a clear justification for the selection of thematic analysis in relation to the study's main objectives. Consider expanding the description of the credibility check to include additional reliability and validity measures like triangulation and member checking.

4. Semi-Structured Interviews sub-section: Provide sufficient detail of the steps followed in creating the interview guide including any theoretical frameworks or literature that shaped the questions. Explain the results of the pilot test for the interview guide and list the changes made to the guide afterward.

5. Provide information about pilot interviews, interviewer training and standardization. These would improve the methodology.

6. Include information about confidentiality measures like data anonymization techniques, and specific protocols followed during transcription and analysis to protect participant identity.

Result Section

The study's results section establishes a strong basis of the findings yet could achieve greater impact through improved clarity and depth of analysis along with better integration of qualitative and quantitative data as well as more comprehensive discussions on participant experiences and technical challenges.

Discussion Section

1. The manuscript would benefit from direct comparisons of findings with existing literature to

2. Consider providing concrete recommendations for tackling the issues found while discussing themes to enhance the insightfulness of the analysis.

Conclusion Section

1. The manuscript recognizes the necessity for additional research but did not clearly outline specific research areas for future exploration.

2. Provide a clearer explanation of how its research findings will influence clinical practice by offering specific suggestions for better integration of rtCGM into young adult type 1 diabetes management strategies.

7. PLOS authors have the option to publish the peer review history of their article (what does this mean? ). If published, this will include your full peer review and any attached files.

**Do you want your identity to be public for this peer review?** For information about this choice, including consent withdrawal, please see our Privacy Policy .

Reviewer #1: No

Reviewer #4: **Yes: ** ADEWUNMI AKINGBOLA

Reviewer #5: **Yes: ** Olukunle O. Akanbi

Reviewer #6: No

---

## [Author Response · Author response to Decision Letter 2]

21 Feb 2025

We thank the reviewer for their helpful comments and suggestions to improve our manuscript. Please find below our responses to the comments highlighted in bold. The new changes in the manuscript have been highlighted in blue.

Reviewer #6: PEER REVIEW REPORT

Abstract Section

1. The abstract successfully introduces the relevance of rtCGM for type 1 diabetes care in a particular population but does not provide a defined explanation of the knowledge gap it intends to address regarding psychosocial effects.

Thank you, we have expanded on this within the abstract as follows:

“Background: Real-time continuous glucose monitoring (rtCGM) is now the standard care for people with type 1 diabetes. However, whilst its impact on glycaemic outcomes is well-documented, its psychosocial effects, particularly in young adults experiencing extreme hyperglycaemia, remain poorly understood.”

2. The abstracts methods section summarizes participant demographics and data collection and analysis methods but does not provide specific information about the study design . Kindly state the study design e.g., "A qualitative study employing semi-structured interviews...").

Thank you, we added this information as requested. The research design and methods section of the abstract now reads:

“A qualitative study employing semi-structured interviews was undertaken. Young adults 18-25 years (HbA1c >75mmol/mol (9.0%)), naïve to rtCGM, were provided with rtCGM for 6-months. Interviews (centred on barriers to self-management and experience of rtCGM use) were conducted within 2-weeks of recruitment and at the end. An inductive, thematic analysis of interviews was undertaken.”

3. The results section: In addition to the primary findings and thematic insights outlined, consider including a brief quantifications of the major outcomes

Thank you for your suggestion. In the abstract, we have focused primarily on qualitative outcomes. We did not draw attention the quantitative data from the study that may be difficult to interpret in light of the small numbers of participants studied. However, we have provided individual HbA1c values within the manuscript to offer context.

4. Consider using structured abstracts formatted with headings such as Background, Objectives, in addition to Methods, Results, and Conclusions already included.

Thank you, we have introduced the subheadings of “Background” and “Objectives” into the abstract in place of “Introduction”.

5. Review the conclusion to ensure it connects the study findings with its objectives and the research gap more effectively.

Thank you. We have added the following statement to the conclusion:

“Implementing structured educational, psychosocial, and technical support, alongside alternative care models such as more frequent check-ins, should be considered in order to enhance self-management practices with rtCGM and address technology-related challenges.”

Introduction Section

1. The introduction establishes a solid foundation through its discussion of intensive self-management for type 1 diabetes and the challenges faced by young adults. Consider providing explicit definition of the research gap concerning the psychosocial effects of rtCGM for high-risk group.

Thank you – we have expanded on the following paragraph, highlighting this is a clear research gap.

“There is limited qualitative research looking into the views and experiences of rtCGM usage in high-risk young adults with type 1 diabetes and there is a research gap concerning the psychosocial effects of rtCGM for this high-risk group. It is therefore unknown how rtCGM may affect thoughts and behaviours that are barriers to achieving optimum or safe glycaemia in this group. There is a clinical need to identify strategies that can address barriers in this vulnerable group.”

2. The introduction could be strengthened by explicitly specifying the research questions or hypotheses.

Thank you. The research questions have been added to the end of the introduction, which now reads:

This study addresses two key research questions: (1) How do individuals with high-risk T1D experience and manage life with diabetes, including barriers to self-management? (2) How do participants perceive and use rtCGM, including challenges and its impact on diabetes management?”

Method Section

1. Participant sub-section: Provide more comprehensive detail of the recruitment process. Briefly discuss challenges encountered during recruitment such as participants willingness and recruitment rate.

Thank you, we have expanded further on this as follows:

“Participants were recruited sequentially from the young-adult and general type 1 diabetes clinics at Imperial College Healthcare NHS Trust (ICHNT), UK, between 20 November 2019 and 22 July 2021. Recruitment was challenged by the COVID pandemic but in general participants were keen to take part.”

2. Although you mentioned that participant were recruited from Imperial College Healthcare NHS Trust diabetes clinics in the UK . Kindly provide information on how the clinic was selected,.

Thank you, we have now specified these were type 1 diabetes clinics, particularly the dedicated young adult type 1 diabetes clinic.

3. Analysis sub-section: Provide a clear justification for the selection of thematic analysis in relation to the study's main objectives. Consider expanding the description of the credibility check to include additional reliability and validity measures like triangulation and member checking.

We have now included an additional sentence in the analysis section: “Pseudonymized transcripts were analysed using thematic analysis (22), which was selected due to its flexible theoretical methodology and rigour.”

We have additionally specified: “Credibility checks (23) involved a second researcher examining three analyzed interview transcripts to provide feedback on codes (PA), and further reviewing themes and subthemes, both individually, and together as a group (VM, PA, SM).”

4. Semi-Structured Interviews sub-section: Provide sufficient detail of the steps followed in creating the interview guide including any theoretical frameworks or literature that shaped the questions. Explain the results of the pilot test for the interview guide and list the changes made to the guide afterward.

Thank you, we have added the following to the semi-structured interview section of the manuscript: “The interview schedule was developed by the research team.”

We have also included credibility checks: “Credibility checks involved a second researcher examining three analysed interview transcripts to provide feedback on codes (PA), and further reviewing themes and subthemes, both individually, and together as a group (VM, PA, SM).”

As the study employed inductive thematic analysis, the interview guide was exploratory rather than based on pre-existing theories or frameworks. While the interviews were not piloted—a common practice in qualitative research—the questions were informed by existing literature on barriers to self-management and developed by researchers with expertise in diabetes care and extensive experience in patient discussions.

5. Provide information about pilot interviews, interviewer training and standardization. These would improve the methodology.

Thank you, all interviews were conducted by a diabetes specialist clinical psychologist. We have added that the psychologist was experienced in conducting qualitative research interviews and this sentence now reads:

“All participants were invited to take part in two semi-structured interviews with a Diabetes Specialist Clinical Psychologist experienced in conducting qualitative research interviews, one within two weeks of enrolment (baseline), and another after six months of rtCGM.”

6. Include information about confidentiality measures like data anonymization techniques, and specific protocols followed during transcription and analysis to protect participant identity.

Thank you, we have added the following to the manuscript:

“To ensure confidentiality, all transcripts were pseudonymized by replacing identifying details with unique participant codes. Data were stored securely on encrypted devices, and access was restricted to authorised researchers. During transcription, any potentially identifiable information was removed or altered to maintain participant anonymity, following established ethical guidelines.”

Result Section

The study's results section establishes a strong basis of the findings yet could achieve greater impact through improved clarity and depth of analysis along with better integration of qualitative and quantitative data as well as more comprehensive discussions on participant experiences and technical challenges.

Thank you for the recognition of the strong basis of findings in the results section. We do have quantitative data presented in Tables 2 and 3, which provide context for individual participants, including sensor wear time and HbA1c change. Followed by addressing this within the manuscript. For example, we highlight that the five participants who reported technical difficulties with rtCGM (Participants 4, 5, 6, 7, and 8) had the lowest sensor usage (all under 40% wear time), with some indicating that they would have worn it more had these challenges been resolved. However, as mentioned above in our responses, due to small participant numbers, we did not attempt to summarise or draw conclusions from the quantitative findings.

Discussion Section

1. The manuscript would benefit from direct comparisons of findings with existing literature to

Thank you for your feedback. We have already included comparisons where relevant, however as this is an under-studied group there are a limited number of studies to directly compare to. We highlight paragraphs 2-4 in the discussion section which cite the existing literature to frame the findings of our study in context.

2. Consider providing concrete recommendations for tackling the issues found while discussing themes to enhance the insightfulness of the analysis.

Thank you – we have rephrased the following section as follows:

“We recommend this cohort to benefit from additional support in using such technologies with more intensive training and psychosocial support. As there is a shift towards more virtual rtCGM initiations or indeed, self-starts using online educational videos, it is important to consider that younger adults may need more intensive support when starting rtCGM than may be assumed.”

Conclusion Section

1. The manuscript recognizes the necessity for additional research but did not clearly outline specific research areas for future exploration.

Thank you – the following statement is included within our manuscript, and we have now clearly outlined which specific populations to be considered.

“Further research is required on how to engage and support young adults in the highest risk groups (HbA1c > 75 mmol/mol (9%) or ≥ 1 DKA admissions or ≥ 1 hospital admission with extreme hyperglycaemia) with rtCGM, and whether a more structured educational and psychosocial support package with greater contact time than is the current standard of care, and specific technical support, may offset challenges from technology usage.”

2. Provide a clearer explanation of how its research findings will influence clinical practice by offering specific suggestions for better integration of rtCGM into young adult type 1 diabetes management strategies.

We have added a following statement to the paragraph above:

“These research findings will influence clinical practice through highlighting the need for better integration of rtCGM into young adult T1D management through alternative care models, such as more frequent check-ins and targeted interventions for high-risk groups.”

---

## [Editor Report · Decision Letter 2]

23 Feb 2025

The experiences of high-risk young adults with type 1 diabetes transitioning to real-time continuous glucose monitoring – A thematic analysis

PONE-D-24-45438R2

Dear Dr. Misra,

We’re pleased to inform you that your manuscript has been judged scientifically suitable for publication and will be formally accepted for publication once it meets all outstanding technical requirements.

Kind regards,

Yusuf Oloruntoyin Ayipo, Ph.D

Academic Editor

PLOS ONE

Additional Editor Comments (optional):

The study is well-designed and the manuscript has been composed in ideal format. The authors have responded appropriately to all concerns raised by the reviewers and I believe the quality of the submission matches the standards for publication in this journal.
---

## [Editor Report · Acceptance letter]

PONE-D-24-45438R2

PLOS ONE

Dear Dr. Misra,

I'm pleased to inform you that your manuscript has been deemed suitable for publication in PLOS ONE. Congratulations! Your manuscript is now being handed over to our production team.

Kind regards,

on behalf of

Dr. Yusuf Oloruntoyin Ayipo

Academic Editor

PLOS ONE